

# EUREC[4]A's HALO

Heike Konow[1,2], Florian Ewald[3], Geet George[2,4], Marek Jacob[5], Marcus Klingebiel[6,2], Tobias Kölling[2,7], Anna E. Luebke[6], Theresa Mieslinger[1,4], Veronika Pörtge[7], Jule Radtke[1,4], Michael Schäfer[6], Hauke Schulz[2,4], Raphaela Vogel[8], Martin Wirth[3], Sandrine Bony[8], Susanne Crewell[5], André Ehrlich[6], Linda Forster[7], Andreas Giez[9], Felix Gödde[7], Silke Groß[3], Manuel Gutleben[3,7], Martin Hagen[3], Lutz Hirsch[2], Friedhelm Jansen[2], Theresa Lang[1,4], Bernhard Mayer[7], Mario Mech[5], Marc Prange[1,4], Sabrina Schnitt[5], Jessica Vial[8], Andreas Walbröl[5], Manfred Wendisch[6], Kevin Wolf[6,10], Tobias Zinner[7], Martin Zöger[9], Felix Ament[1,2], and Bjorn Stevens[2]

[1]Universität Hamburg, Hamburg, Germany
[2]Max Planck Institute for Meteorology, Hamburg, Germany
[3]Deutsches Zentrum für Luft- und Raumfahrt, Institut für Physik der Atmosphäre, Oberpfaffenhofen, Germany
[4]International Max Planck Research School on Earth System Modelling, Max Planck Institute for Meteorology, Hamburg, Germany
[5]Institute for Geophysics and Meteorology, University of Cologne, Cologne, Germany
[6]Leipzig Institute for Meteorology, University of Leipzig, Leipzig, Germany
[7]Ludwig Maximilians Universität, Meteorologisches Institut, Munich, Germany
[8]LMD/IPSL, CNRS, Sorbonne University, Paris, France
[9]Deutsches Zentrum für Luft- und Raumfahrt, Flugexperimente, Oberpfaffenhofen, Germany
[10]Laboratory for Atmospheric and Space Physics, University of Colorado Boulder, Boulder, Colorado

**Correspondence:** Heike Konow (heike.konow@mpimet.mpg.de)

**Abstract.** As part of the EUREC[4]A field campaign, the German research aircraft HALO, configured as a cloud observatory, conducted 15 research flights in the trade wind region east of Barbados in January and February 2020. Narrative text, aircraft state data, and meta data describing HALO's operation during the campaign are provided. Each HALO research flight is segmented by time-stamp intervals into standard elements to aid the consistent analysis of the flight data. Photographs from

HALO's cabin and animated satellite images synchronized with flight tracks are provided to visually document flight conditions. As a comprehensive product from the remote sensing observations, a multi-sensor cloud mask product is derived and quantifies the incidence of clouds observed during the flights. In addition, to lower the threshold for new users of HALO's data, a collection of use cases is compiled into an online book "How to EUREC[4]A", included as an asset with this paper. This online book provides easy access to most of EUREC[4]A's HALO data through an intake catalogue.



## 1 Introduction

The EUREC$^4$A (ElUcidating the RolE of Cloud–Circulation Coupling in ClimAte, Bony et al., 2017) field campaign took advantage of the capabilities of the cloud-observatory configuration of the German research aircraft HALO (High Altitude and Long-range Research Aircraft, Krautstrunk and Giez, 2012). This configuration, as described by Stevens et al. (2019),

was developed and implemented over the course of several previous HALO campaigns, two of which – NARVAL-South and NARVAL2 (Next generation Advanced Remote sensing for VALidation Studies) based out of Barbados – were in direct preparation for EUREC$^4$A. As motivated by Bony et al. (2017) and described by Stevens et al. (2021), EUREC$^4$A made measurements to (i) test hypothesized mechanisms that would cause large reductions in trade-wind cloudiness with warming; and (ii) to benchmark a new generation of global storm-resolving models (Satoh et al., 2019).

HALO was one of four scientific platforms forming the nucleus of EUREC$^4$A. Its measurements were closely coordinated with those from the other three core platforms – the research vessel (R/V) Meteor, the Barbados Cloud Observatory (BCO, Stevens et al., 2016), and the French SAFIRE ATR-42. Two additional aircraft, three further research vessels and a small fleet of air- and water-borne robotic instrument platforms supported a substantial broadening of EUREC$^4$A's initial scope and, as described by Stevens et al. (2021), involved looser coordination with HALO. In this manuscript we elaborate on HALO's

contribution to EUREC$^4$A, independent of the other platforms. We do so by by describing how HALO was tasked during EUREC$^4$A, both in standard narrative form, as well as through the provision of auxiliary data and meta data, including flight segmentation data, animated geostationary satellite data with flight tracks, and curated photographs (Sect. 2). Through the provision of aircraft state information and the construction of a multi-sensor cloud mask product, Sect. 3 gives a synthetic overview of HALO's scientific payload, and the varying cloud conditions it observed. Section 4 outlines how to access and use

the HALO measurements as part of a developing data concept. Links to the data and a brief summary are provided in Sect. 5.

## 2 HALO during EUREC$^4$A

HALO is a Gulfstream 550 that has been modified for atmospheric research and is operated by the German Aerospace Center (Krautstrunk and Giez, 2012; Wendisch et al., 2016). During EUREC$^4$A HALO was flown in a slightly updated version of the cloud-observatory configuration described by Stevens et al. (2019). In addition to housekeeping data (aircraft state and

in-situ meteorological measurements), this updated configuration consists of a nadir looking differential absorption and high spectral resolution lidar (Wirth et al., 2009), cloud radar and microwave radiometer (Mech et al., 2014), a zenith oriented spectral radiometer (Wendisch et al., 2001), an imaging spectrometer (Ewald et al., 2016), a thermal imaging polarimeter, an infrared imager, a dropsonde system, and broadband radiometers. The imaging polarimeter, infrared imager, and broadband radiometers were new additions to the HALO cloud-observatory configuration. In this section we describe how and where

HALO was deployed. This description is aided by the development of a meta data concept (and the meta data arising from its application) to systematically segment the flight data and document the meteorological conditions (through photographs and satellite imagery) encountered on the different flights.



## 2.1 Flights

**Table 1.** HALO research flights during EUREC⁴A. Except for HALO-0119 and HALO-0218, all flights were local flights in that they took off and landed at Grantley Adams International Airport on Barbados. All times given as UTC. The special features column gives information about the purpose of each flight aside from the EUREC⁴A-Circle pattern.

| Flight ID | Date | Take-off | Landing | Duration (h:mm) | Dropsondes | Comment | Special features |
|---|---|---|---|---|---|---|---|
| HALO-0119 | 2020-01-19 | 0934* | 1848 | 9:13 | 14 | Silke's Coming | Transfer to Barbados |
| HALO-0122 | 2020-01-22 | 1457 | 0010 | 9:12 | 70 | Fish Wake | Instrument calibration |
| HALO-0124 | 2020-01-24 | 0929 | 1841 | 9:11 | 75 | ColdPools | Characterizing upstream flow |
| HALO-0126 | 2020-01-26 | 1205 | 2120 | 9:15 | 71 | Manfred's Escape | Aircraft coordination and ship coordination |
| HALO-0128 | 2020-01-28 | 1458 | 2355 | 8:56 | 71 | Sugar | Characterizing upstream flow |
| HALO-0130 | 2020-01-30 | 1119 | 1508 | 3:48 | 4 | Mario's Snail | ATR colocation, GPM underpass |
| HALO-0131 | 2020-01-31 | 1508 | 2356 | 8:48 | 74 | Grains for Geet | Characterizing upstream flow |
| HALO-0202 | 2020-02-02 | 1128 | 2013 | 8:45 | 89 | Felix's Clover | Clover pattern for vertical motion calculation |
| HALO-0205 | 2020-02-05 | 0915 | 1821 | 9:05 | 76 | Bernhard's Bicycle | Terra underpass |
| HALO-0207 | 2020-02-07 | 1202 | 2111 | 9:09 | 73 | Raphaela's Flower | Characterizing upstream flow |
| HALO-0209 | 2020-02-09 | 0914 | 1803 | 8:48 | 72 | Sabrina's Towers | Characterizing upstream flow |
| HALO-0211 | 2020-02-11 | 1229 | 2137 | 9:08 | 61 | Marek's Intermezzo | GPM underpass |
| HALO-0213 | 2020-02-13 | 0756 | 1717 | 9:21 | 73 | Jessica's Veils | Characterizing upstream flow |
| HALO-0215 | 2020-02-15 | 1507 | 0012 | 9:05 | 50 | Under Cover | Above and below altostratus layer |
| HALO-0218 | 2020-02-18 | 1011 | 1855† | 8:44 | 7 | Silke's Going | Transfer from Barbados |

* Take-off at Santiago de Compostela Airport, Spain; † Landing at Oberpfaffenhofen Airport, Germany

HALO performed fifteen research flights on fifteen different days in support of EUREC⁴A, as listed (with an evocative moniker) in Table 1. Flight IDs in the format HALO-MMDD, rather than an enumeration of the research flights, are used to distinguish the different flights. This helps avoid confusion arising from non-coincident flights among the various research aircraft contributing to EUREC⁴A. Thirteen of these (HALO-0122 to HALO-0215) are designated as local flights, as they had both the take off and landing at Barbados' Grantley Adams International Airport. With the exception of HALO-0130 – a short flight that took advantage of overlap in crew duty to make some additional measurements of opportunity – each local flight lasted about 9 h, with roughly 7 h of circling on what Stevens et al. (2021) call the 'EUREC⁴A-Circle.' This circle was

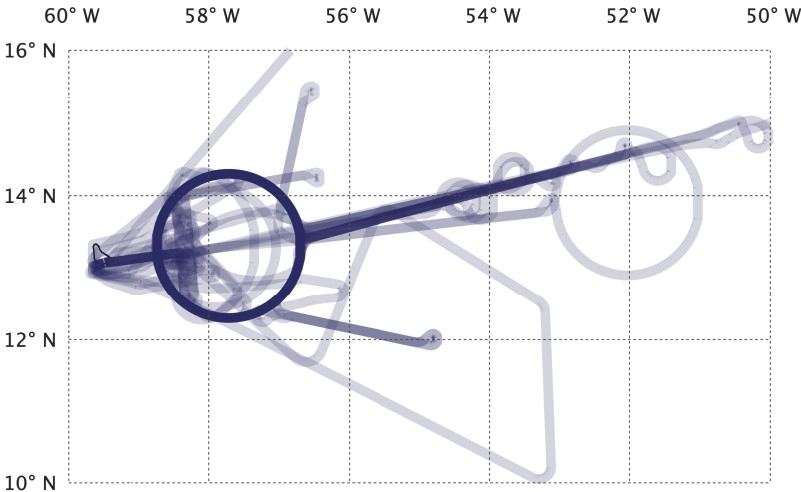

**Figure 1.** Heat map of HALO flight tracks from all 15 flights. The darkness of the color represents the frequency a location was visited. Map data based on Wessel and Smith (1996).

largely defined by the HALO flight pattern, which was fixed before the beginning of the campaign to support the deployment of dropsondes around a geographically fixed circle positioned windward of the BCO, far enough upwind to not interfere with commercial air traffic, but not so far as to be out of range of a C-band polarized research radar (POLDIRAD). The EUREC[4]A-Circle is easily identified as the darkest circle area in the heat map of flight tracks in Fig. 1, with a center at 13.3° N, 57.717° W and an approximately 220 km diameter.

An important and unusual aspect of the HALO (and EUREC[4]A) flight strategy was that it did not target specific meteorological conditions. Flight days were scheduled in coordination with the ATR so as to maximize the utilization of the aircraft subject to crew duty restrictions. Variations in take-off (and landing) times were implemented to better sample the diurnal cycle, and staggered to accommodate crew-duty considerations, rather than to target specific meteorological conditions. On most flights some time was also dedicated to flight elements other than the EUREC[4]A-Circle, for instance to allow an underpass of a meteorological satellite (e.g., the Terra satellite during 'Bernhard's Bicycle'), or to sample the upwind conditions that were being monitored by other platforms. Only 'Mario's Snail' (HALO-0130), the south-east excursion on 'Manfred's Escape' (HALO-0126), which coordinated sampling of a cirrus deck with the R/V Meteor, and the choice of flight levels on 'Under Cover' were influenced by meteorological conditions. The moniker associated with each flight (Table 1) was chosen to strengthen the mental image associated with that flight, and in most cases remind the reader of the principle investigator (PI) of each flight.

## 2.2 Flight segmentation

To aid in the analysis of flight data, all HALO flights are segmented via timestamps into a system of hierarchical identifiers. Non-exclusive segments are defined by two '(YYYY-MM-DD hh:mm:ss)' timestamps, the first one defining the start of the

segment, the second denoting the first time after the end of the segment. Timestamps have a temporal resolution of 1 s and times
are given in UTC. Every segment belongs to a "kind" - a categorical type for segments defined in Table 2. It helps to think of
segments as an interval of flight time and the corresponding "kinds" as describing how the aircraft was being operated during
this time interval (Fig. 2). Flight segmentation data is provided as YAML (YAML Ain't Markup Language) files that can be

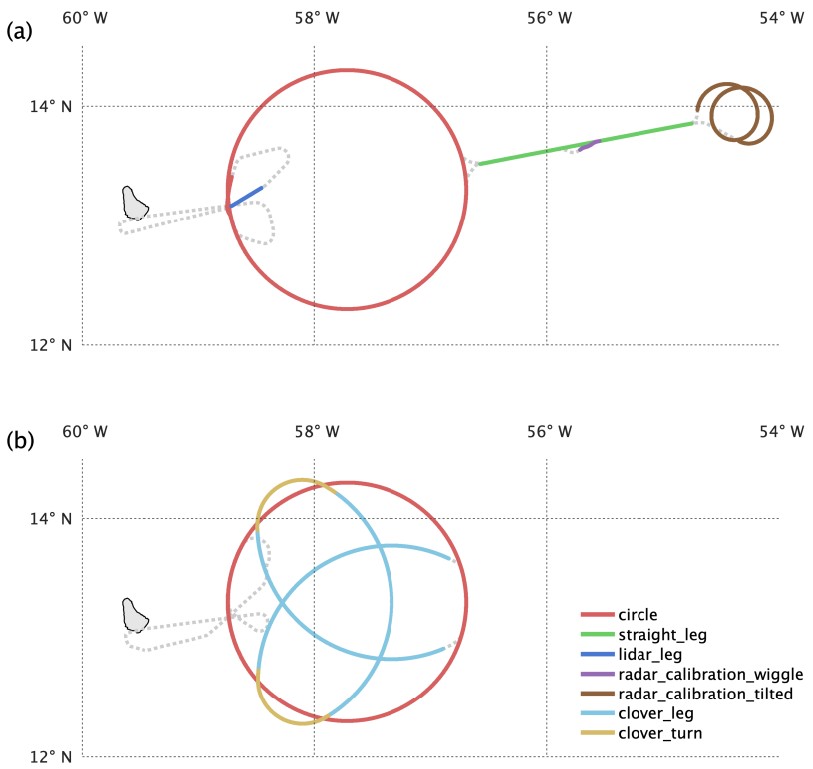

**Figure 2.** Examples of flight segments (colored) for two research flights. a) on flight HALO-0131, b) on flight HALO-0202. Portions of flight
track that are not segmented appear as dotted lines. Map data based on Wessel and Smith (1996).

accessed at https://doi.org/10.5281/zenodo.4900003. This section provides a description of the YAML files and the reasoning
behind their structure and method.

By adopting non-exclusive segments, a timestamp can belong to multiple segments that differ in kind. For example, times-
tamps belonging to the kind 'lidar leg' will also belong to the kind 'straight leg', if they match the definition of the latter.
Segment start and end times were first roughly categorised based on timestamps from the flight reports and aircraft navigation
features such as roll angle, altitude, pitch, etc. However, the final attribution of timestamps to segments were performed man-
ually by the listed 'contact' in the YAML files. At least one other person later tested the segmentation for errors or avoidable
deviations from the kind definitions. Because the segmentation was performed manually, segments are defined by the time
intervals that are assigned to them, rather than by their kind.



**Table 2.** Definition of flight segments. The total number of these segments identified from all flights has been provided in the right-most column.

| Segment | Description | Total |
|---|---|---|
| circle | Circles are based on a set of twelve dropsondes. Circle starts 1 min before the first launch and ends 360° later without overlap. This describes a roughly 1 h flight segment along a circular path at constant altitude, with roughly 2° roll angle, and a start and end point within 30° of one another, as defined by radials from the circle center. | 72 |
| circle break | Periods between two consecutive circles, during which no dropsondes were launched. It is assured that the aircraft remained on the circle track. Circle breaks may be used to obtain all the available remote sensing data from circles, neglecting availability of dropsonde data. | 45 |
| circling | Period during which the aircraft was on the standard circling track with roughly 2° roll angle. Periods without dropsonde launches are included here (e.g. circle break). Useful when wanting to loop over the full period HALO was on the circle track. | 26 |
| straight leg | Period with constant aircraft heading, altitude and close to 0° roll angle (max. 3° roll for short periods). Straight legs were flown with various purposes, which are more closely described by the straight leg "name"-parameter in the YAML files and are in some cases also expressed by additional entries in the segment "kinds" attribute. | 51 |
| lidar leg | Maneuver typically conducted at flight level (FL) 160 along the return ferry of each local research flight. Defined as the period of the aircraft being in FL160. If roll angle was close to 0° the whole time, the segment is also of kind "straight leg". | 12 |
| radar calibration wiggle | Maneuver typically conducted during straight legs, where the aircraft rolls with constant roll rate between $\pm 20°$. If conducted during a straight leg, the straight leg is split into three flight segments: 1.) straight leg, 2.) radar calibration wiggle, 3.) straight leg. Segments start and end at about 0° roll angle. | 11 |
| radar calibration tilted | Maneuver typically conducted at the end of a straight leg, where a narrow circle pattern with a constant 10° bank is flown. A constant roll angle of about 10° is used to define the period of a this segment. | 6 |
| baccardi calibration | Defined by 4 turns of 90° indicated by roll angles of about 25° (1 turn: -25°, 3 turns: +25°) to calibrate the BACARDI instrument (Tab. 4). | 1 |
| clover leg | Defined as the long legs of a clover flight pattern with close to 2° roll angle. Dropsondes were launched every 30° along clover legs. The transitions between circle pattern and clover pattern are excluded, because of steep roll angles of about 30°. Clover legs are not defined via launch times of first and last dropsonde, because dropsondes do not always represent the whole leg. | 3 |
| clover turn | Periods between two consecutive clover legs (smooth transition), with steeper roll angles of about 6°. These periods are constrained to the periods during the clover pattern where the aircraft roll angle deviates clearly from 2°. During these turns no dropsondes were launched. | 2 |



Segments also contain a field called 'dropsondes', which provides a list of the dropsondes, whose time of launch are associated with the respective segment. The dropsondes are provided with classifications of *good*, *bad* and *ugly*, based on their

QC classification types from the EUREC[4]A dropsonde dataset, JOANNE (Joint dropsonde-Observations of the Atmosphere in tropical North atlaNtic meso-scale Environments, George et al., 2021). The list is in the form of unique dropsonde IDs that correspond to the variable 'sonde_id' provided in JOANNE and are the 'cf_role' variable therein. This field makes it convenient for selection of the dropsondes based on flight segments. In a few instances the launch time of a dropsonde will fall outside of the segment with which it is associated – for instance if the last sonde of a circle was inadvertently launched too late, after

HALO had completed a circle.

**Table 3.** List of standard irregularities attributed to flight segments

| Keyword | Description |
| --- | --- |
| TTFS | Time To First Sonde: For circle segments, when the start time is set to less than one minute before the launch time of the first sonde in the circle, this tag is attached. |
| SAM | Sonde Attributed Manually: For circle segments, when certain sondes are manually attributed and not as per launch time and segment times. This irregularity is added to the segment and the respective sonde ID is noted. A dropsonde tagged as SAM is attributed manually to the segment which originally fits the dropsonde's planned purpose, e.g. a dropsonde launched as part of the 12-sonde set of a circle, but its location exceeded the 360° point of the circle and therefore its launch time is later than the circle's end timestamp. |
| NONSTD | NON-STanDard segment: Used for circle segments which do not conform to standard EUREC[4]A-Circle features. If a flown circle had a different diameter or a different center location than the EUREC[4]A-Circle, then this tag is used. |

Flight segments that deviated from the allowed kinds, are flagged by an irregularity field. For instance, the inclusion of sondes launched before or after its associated flight segment constitutes an irregularity and is marked. The irregularity field takes the form of an explanatory string describing the irregularity. As the segmentation process revealed some oft-repeated irregularities, standardized irregularity tags (keywords) were defined (Table 3) and are prepended to the explanatory string of

the irregularity field when applicable.

In total 220 segments were defined over the fifteen flights. These included 72 circles (69 regular, one with smaller diameter, one outside of the EUREC[4]A-Circle, and another without dropsonde launches) within 26 periods of circling. Fifty-one straight legs were flown. The segmentation data is published by Prange et al. (2021).

## 2.3  Satellite movies

To give further insight into the large-scale conditions of each flight, satellite movies overlaid with the time evolving flight tracks are created. Snapshots of these movies are shown in Fig. 3 for each flight of HALO. The snapshots were chosen to capture the cloud scene roughly 3 h after take-off. Like the snapshots, the actual movies (Schulz et al., 2021) are based on the 1 min





meso-scans of the Advanced Baseline Imager (ABI) on-board the GOES-16 satellite (GOES-R Calibration Working Group and GOES-R Series Program, 2017), when these are available. During daytime reflectance (channel 2; 0.64 μm) and during
nighttime brightness temperature (channel 13; 10.35 μm) are used. On a few days the ABI did not provide meso-scans over the EUREC⁴A domain. In these cases, 10 min full-disk scans were substituted. To foster the generation of movies with different overlays by users, the source code is available (Fildier et al., 2021) and relies purely on publicly available data sources.

## 2.4 Photographs

During all research flights photographs were taken to visually document the conditions being sampled (Fig. 4). Most photos
were taken by the principle investigator, through either the left or right window in the middle of the cabin forward of the wing, a few were taken from the cockpit. A subset (50–100 per flight) of these photos have been selected and further curated as described below. These photos, with their extended meta data, are included as part of EUREC⁴A's HALO dataset.

The data curation involved manually correcting camera time-stamps by calibrating the camera's internal clock with photographic evidence of flight-level time data from GPS watches or instrument panels synchronised with the aircraft sensor system
time (BAHAMAS, Sec. 3.1). GPS location and altitude tags are added to each photo using BAHAMAS location data at the capture time. For photographs taken on the apron, where aircraft position data is not available, the position of the usual parking position (13.08° N, 59.4828° W) was used. With a cruising air speed of 200 m s⁻¹, the estimated 1 min accuracy of the capture time implies a GPS location accuracy of about 12 km.

Additional meta data was added using standard IPTC (International Press Telecommunications Council) meta data conven-
tions. The IPTC tag "description" is used to describe the scene photographed. The IPTC tag "keywords" contains information about the orientation (viewing direction), the platform HALO, pictured cloud types or other notable objects. In cases where the orientation could not be determined a default is adopted, usually to the left or right of the PI seat. Because most of the photos were taken with a shared camera, some may have been taken by different members of the flight crew; when this was not documented, the PI of each flight is set as the Creator. The supplementary photo documentation is written into each photo's
IPTC tags as part of its extended meta data. The photographs can be viewed and downloaded from the database (Konow et al., 2021).

## 3 Instrumentation

In this section we describe data compiled and published to document HALO's state, as well as the cloud conditions sampled by its different cloud-sensitive instruments. With the exception of the dropsondes, these data are derived from, and thus introduce,
the full suite of instrumentation (Tab. 4) included as part of the cloud-observatory configuration of HALO. Information on to how to access the actual measurements from HALO's instrumental payload, some of which are independently published, is provided in Sect. 4.

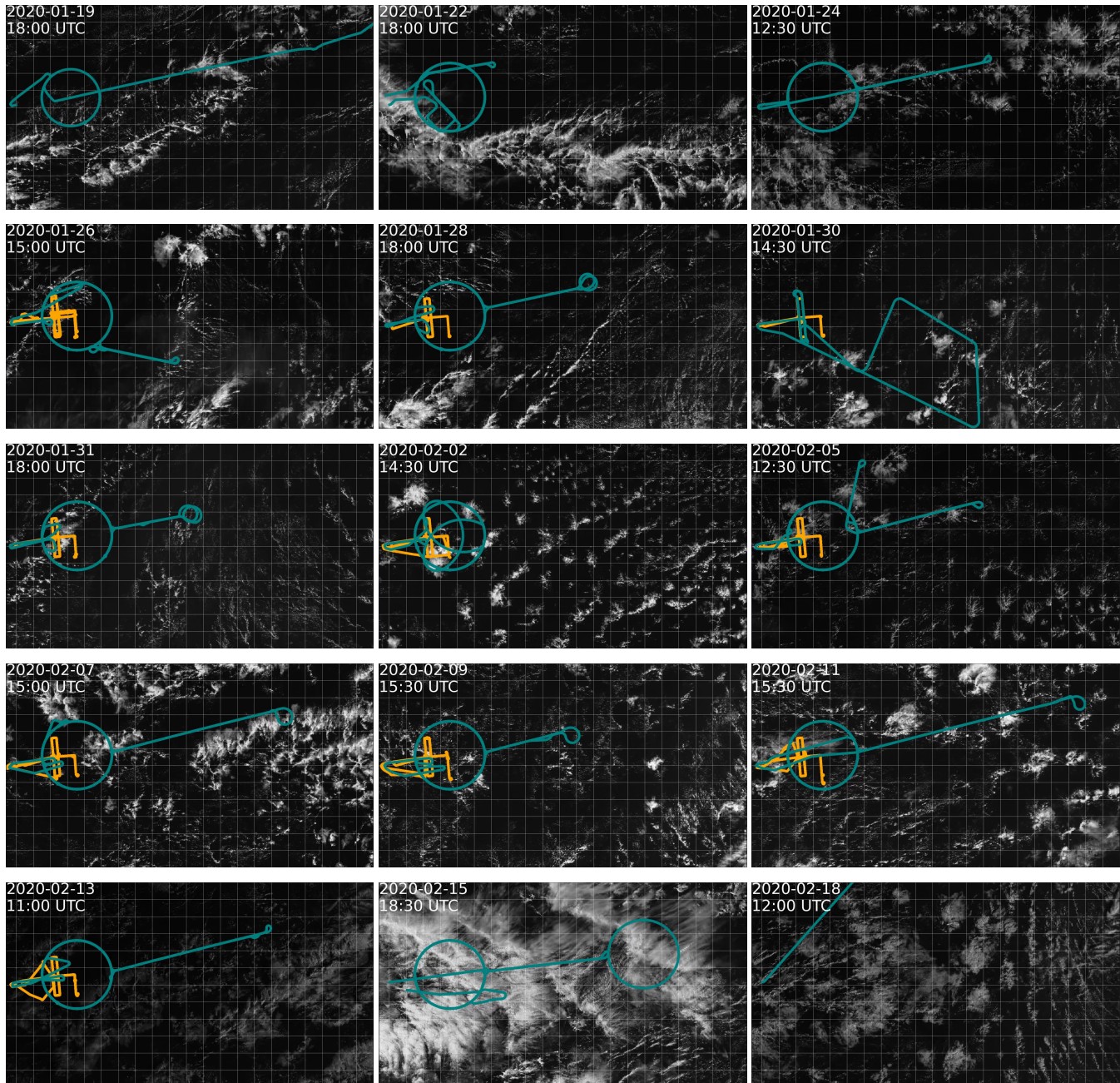

**Figure 3.** Snapshots of animations of GOES-16 ABI images (channel 2; 0.64 μm) for all flights. Tracks of the HALO and ATR aircraft are indicated in teal and orange, respectively. Snapshots are from about mid-flight time of HALO, except for the ferry flights.



**Table 4.** Instrument specifications

| Instrument | Description |
| --- | --- |
| WALES | The water vapor differential absorption lidar WALES (WAter vapor Lidar Experiment in Space, Wirth et al., 2009) operates at four wavelengths in the 935 nm $H_2O$ absorption band for the measurement of water vapor. Additional channels at 532 nm and 1064 nm provide backscatter ratio, and aerosol depolarisation ratio. At 532 nm, an additional High Spectral Resolution Lidar (HSRL) channel allows the retrieval of the atmospheric transmission without assumptions about the extinction to backscatter ratio of aerosol and cloud particles. |
| HAMP | The HALO Microwave Package (HAMP, Mech et al., 2014) is a combination of active and passive sensors in the microwave part of the spectrum. The polarimetric Ka-band MIRA-35 cloud radar provides profiles of the Doppler reflectivity spectrum. Three radiometer modules operate at 25 channels in the range between 20 GHz and 183 GHz. The measurements provide integrated quantities of humidity and liquid water. |
| SMART | The Spectral Modular Airborne Radiation measurement sysTem (SMART, Wendisch et al., 2016; Stevens et al., 2019) measures spectral downward solar irradiances in the wavelength range between 300 nm and 2500 nm. |
| specMACS | The spectrometer of the Munich Aerosol Cloud Scanner (specMACS, Ewald et al., 2016) measures spectrally and angularly resolved radiance in the visible and near-infrared (VNIR camera: 400 nm to 1000 nm; SWIR camera: 1000 nm to 2500 nm) with an up to $35.5°$ wide swath in the across flight track direction. These hyperspectral line imagers were complemented by two polarization resolving RGB cameras with a very large combined field-of-view of about $82°$ in along track and $110°$ in across track direction. |
| BACARDI | The Broadband AirCrAft RaDiometer Instrumentation (BACARDI) is a new radiometer package measuring the downward and upward irradiances at flight level in both the solar (0.2 µm to 3.6 µm) and terrestrial (4.5 µm to 42 µm) wavelength ranges with sets of pyranometers and pyrgeometers, respectively. |
| VELOX | The Video airbornE Longwave Observations with siX channels (VELOX, Schäfer et al., 2021c) thermal infrared camera system comprises the VELOX 327k eL thermal infrared imager operating in the atmospheric window with six spectral channels within the 7.7 µm to 12 µm wavelength range and an infrared pyrometer (KT 19.85 II) measuring in the 9.6 µm to 11.5 µm wavelength range. Two-dimensional fields ($35.5°$ by $28.7°$) of the upward radiance are obtained, which can be converted into brightness temperatures for use in cloud and surface property retrievals. |
| BAHAMAS | The BAsic HALO Measurement And Sensor system (BAHAMAS, Krautstrunk and Giez, 2012) is part of the permanent HALO instrumentation. This system provides aircraft attitude and location data, together with in-situ observations of atmospheric quantities at aircraft level (Sec. 3.1). |
| JOANNE | Dropsonde observations (George et al., 2021) provide in-situ profiles of temperature, humidity, pressure, and wind along the sonde trajectory. |

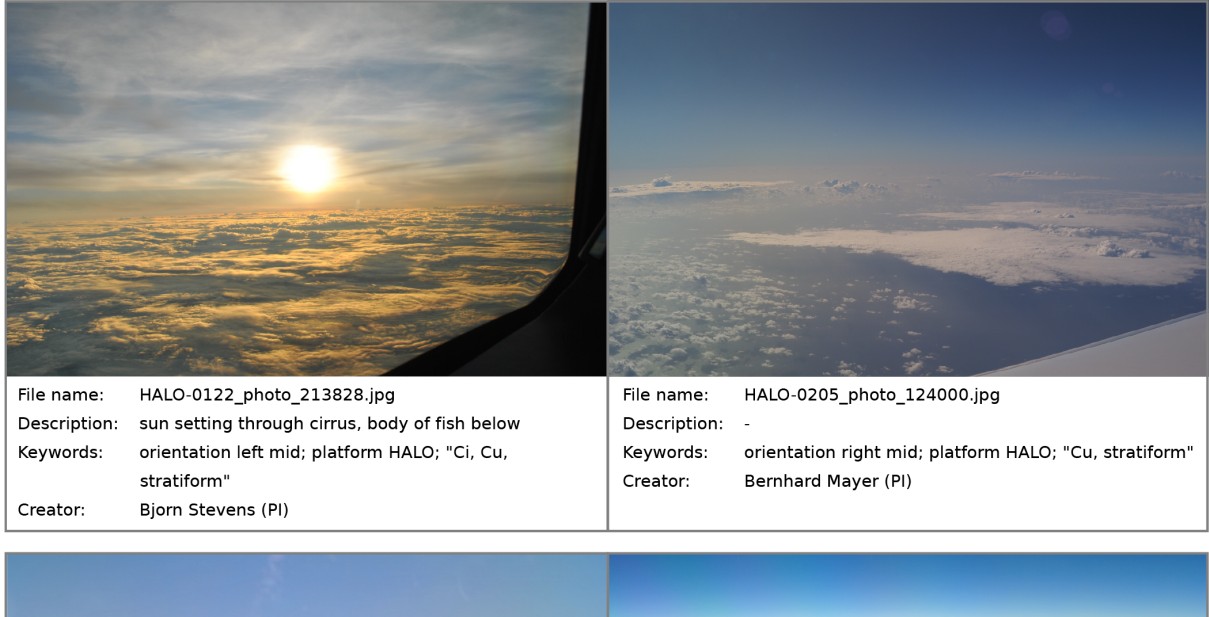

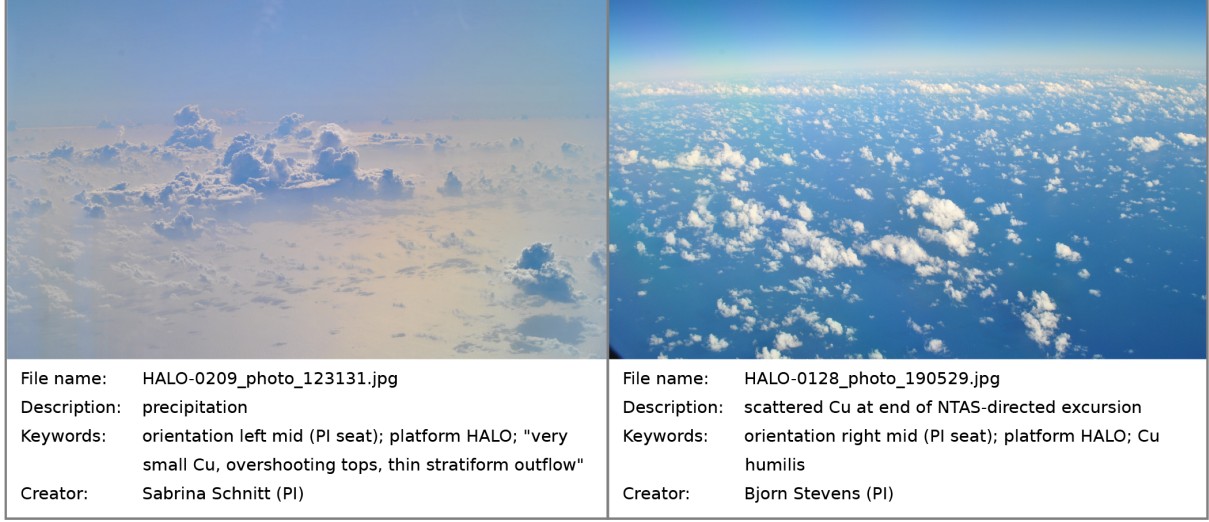

**Figure 4.** Example photographs taken on board HALO with added meta data. The photographs are representative for the Fish, Flowers, Gravel and Sugar type of organization patterns (from top left to bottom right, Stevens et al., 2020)

### 3.1 Aircraft location and attitude data

The Basic Halo Measurement and Sensor System (BAHAMAS, Tab. 4) provides aircraft location and attitude data for all HALO
flights, in addition to atmospheric measurements. A subset of the BAHAMAS data, consisting of aircraft altitude, heading, latitude, longitude, roll angle, pitch angle and true air speed with a time resolution of 10 Hz, has been created (Klingebiel, 2021). Figure 1 uses the data subset to present the tracks of all flights in the vicinity of Barbados as well as the ferry flights from and to Germany. The roll and pitch angle of all flights are shown in Fig. 5. The distribution of the roll angles (blue)



shows two peaks. The one centered at $0°$ indicates straight legs, while the other centered at $2.2°$ arises from circling in a

clockwise (positive roll angle) manner. The distribution of the pitch angle (orange) shows a peak near $3°$. This pitch changes

systematically as fuel is burned through the flight. Although the constant roll angle on the measurements during circling is

sometimes raised as a concern, this analysis shows that – for the large circles flown during EUREC⁴A – the non-zero pitch

results in a larger deviation from true nadir of the downward staring instruments than does the constant roll.

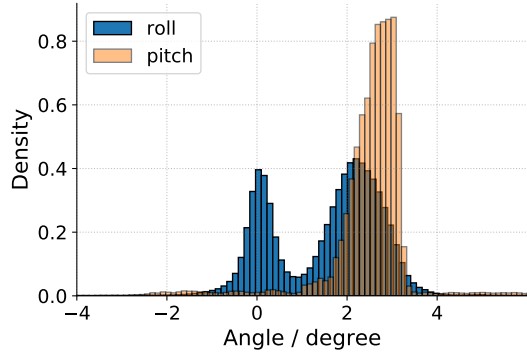

**Figure 5.** Distribution of roll and pitch angles for all HALO flights during EUREC⁴A.

### 3.2    Cloud masks

EUREC⁴A's HALO was designed to observe different properties of clouds using the richness of their interaction with electro-

magnetic radiation. Different instruments (Tab. 4), by virtue of their differing measurement principle and footprint, see clouds

in different ways. Figure 6 provides a snapshot for a five minute flight segment from flight HALO-0205 on a circle segment

(HALO-0205_c2, Tab. 2), which represents typical cloud conditions of EUREC⁴A. WALES and the HAMP radar provide

vertical cross sections, specMACS and VELOX provide a two-dimensional horizontal view of the clouds along the flight path,

and other instruments provide a scalar time-series of measurements along the flight path.

To provide an overview of the cloud fields sampled by HALO, a trinary cloud mask is created for each cloud sensitive

instrument, as described in Appendix B. The access to the cloud mask data is listed in Table 6. The value of the cloud mask

denotes measurements that each instrument identifies as either: *cloud free* (0), *probably cloudy* (1) or *most likely cloudy* (2).

Introduction of the *probably cloudy* reflects the ambiguity in cloud detection faced by many instruments. Especially for the

passive instruments (HAMP radiometer, specMACS, KT19, VELOX), a range of thresholds were applied to separate cloudy

and cloud-free observations. Cases where the lower and upper threshold give a different decision are marked as probably cloudy.

A comparison of the cloud masks (Fig. 6) shows how cloud amount is sensitive to the manner of detecting clouds. The radar is

sensitive to large drops, which form through the collision and coalescence of cloud droplets, a process that becomes active as

clouds deepen and increase their condensate burden. The lidar, on the other hand, is also sensitive to optically thin clouds with

a very small condensate burden. This explains the differences in the measured cloud cover by these two instruments for the

five minute segment shown in Fig. 6. The sensitivity of the passive instruments is influenced by the contrast of the cloud and



**Figure 6.** Example scene of cloud masks from different instruments during research flight HALO-0205. Panel (a) shows the backscatter ratio at 1024 nm from WALES together with a cloud top height estimate. (b) shows the HAMP cloud radar reflectivity, (c) a horizontal view on the cloud field from the specMACS imager at 1.6 μm (SWIR, short wave infrared), and (d) a horizontal view from the VELOX IR imager (7.7 μm and 12 μm). Panel (e) shows a scalar cloud mask product along the flight path from six instruments. The three cloud flag values can be used to derive a minimum or maximum cloud-cover stated on the right. Minimum cloud-cover includes only *most likely cloudy* cases, maximum cloud cover includes *most likely cloudy* and *probably cloudy* cases. For the comparison only the central 11 x 11 pixels (0.57°) from VELOX and central 0.6° from specMACS are selected, both as close as possible to the HAMP cloud radar footprint.



**Table 5.** Campaign mean cloud cover estimates from all local research flights (22 Jan - 15 Feb). Minimum cloud-cover: only *most likely cloudy*, maximum *most likely cloudy* and *probably cloudy* cases. Note that not all instruments performed measurements at all times.

| instrument | cloud cover | |
| --- | --- | --- |
| | minimum | maximum |
| WALES | 0.34 | 0.34 |
| HAMP Radar | 0.21 | 0.22 |
| specMACS | 0.16 | 0.22 |
| HAMP Radiometer | 0.16 | 0.25 |
| KT19 | 0.20 | 0.31 |
| VELOX | 0.21 | 0.39 |

surface reflection or emission. A time offset is also apparent in different cloud flags, which arises from slight differences in the instrument orientations (more forward pointing instruments detect clouds earlier than more backward pointing instruments), rather than lack of synchronicity.

The campaign average cloud-cover estimates as detected by the instruments are stated in Table 5. Most instruments define a minimum cloud-cover based on the cloud flag *most likely cloudy* and a maximum cloud cover that additionally includes the uncertain cloud flag *probably cloudy*. WALES stands out as there is no *probably cloudy* flag in the cloud mask algorithm (Sect. B1), and the minimum and maximum cloud-cover are equal. The HAMP Radar seems to have very few uncertain cases.

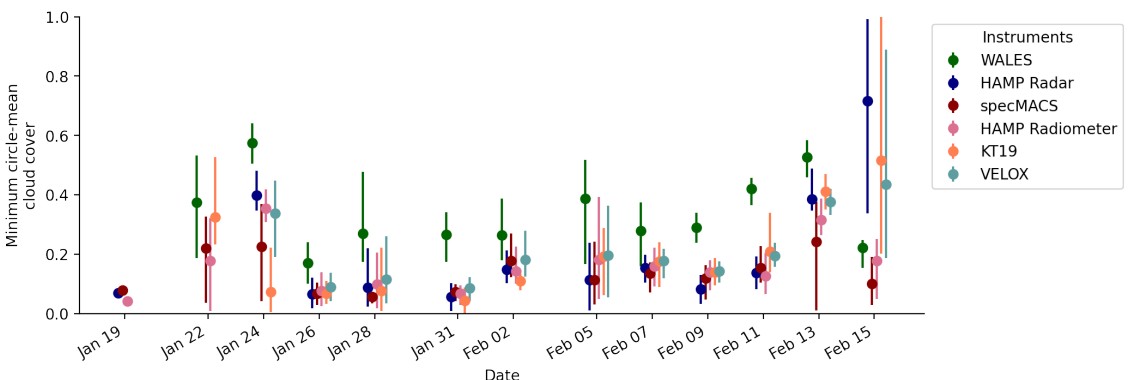

**Figure 7.** Time series of circle-mean (minimum) cloud cover estimates. The markers visualize the research-flight average, while the lines span the range of all circle-mean cloud cover estimates on a respective flight.

To provide context to the variations in cloud cover, Fig. 7 shows a time series of circle-mean (minimum) cloud-cover
estimates for all research flights and from all instruments respectively. HALO typically flew six circles per research flight (per day). In addition to the research flight mean, the whiskers span the range from the smallest to the largest circle-mean



(minimum) cloud cover. For most cases the cloud cover estimates from passive instruments and the radar agree well. WALES systematically detects more clouds. It is more aligned with the circle-mean (maximum) cloud-cover estimates of the other instruments, as it does not include an uncertain cloud flag and is very sensitive to optically thin clouds. The flight HALO-0215
is an exception to the systematic difference between WALES and the other sensors which is due to a deep stratocumulus layer with a strong reflection at cloud top that blinded the lidar, while the radar was still able to provide reasonable estimates. In general, the instrument measurements suggest higher cloud cover in the beginning as well as towards the end of the campaign which agrees with our personal perception.

To further investigate the differences among the sensors and their cloud masking algorithms, we display the cumulative
fraction of circle-mean cloud cover estimates in Fig. 8. In particular, the bars show the range defined by the circle-mean minimum and circle-mean maximum cloud cover estimates for the cloud cover ranges stated on the x-axis. The differences between minimum and maximum cloud cover originate from the uncertain cases with cloud flag *probably cloudy*. The first thing to note is a disagreement between the instruments for cloud cover ranges up to about 0.5 due to their different detection principles. Geometrically and optically thin clouds can have a significant impact on circle-mean estimates in low cloud cover
situations and lead to uncertain pixels depending on the detection principle (Mieslinger et al., submitted). As WALES is able to detect optically thin clouds with few condensates, the cloud cover estimates are generally higher and the change in cumulative fraction is strongest between 0.2 and 0.6. The radar stands in contrast to WALES with most circle measurements exhibiting a cloud cover below 0.2 as it cannot detect the small and optically thin clouds at the operating wavelength. The VELOX cloud mask includes a high fraction of uncertain pixels leading to a large difference (large bars) between the minimum and maximum
cloud cover visible in Fig. 8 at cloud covers up to 0.4. In the case of VELOX as well as for all other passive instruments, the cloud cover estimates shift to higher numbers when the thresholds are reduced (from minimum to maximum cloud cover).

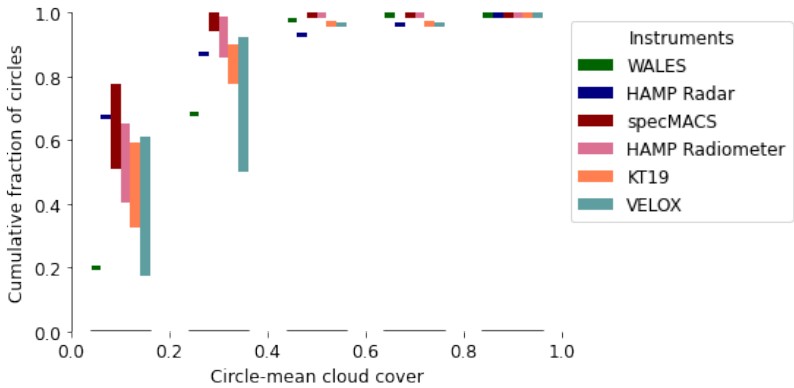

**Figure 8.** Cumulative fraction of circle-mean cloud cover estimates. Depending on the instruments and some instrument downtimes, the available circle counts range from 64 to 72. The bins on the x-axis have a bin width of 0.2 respectively. The bars span the range defined by the minimum cloud cover based on cloud flag *most likely cloudy* and the maximum cloud cover based on cloud flags *most likely cloudy* and *probably cloudy*.





In general we find that only few circles have a cloud cover higher than 0.6. At such high cloud cover the instruments agree remarkably well and also, minimum and maximum cloud cover are almost equal meaning that there are few or none *probably cloudy* measurements. Viewed differently, about 90% of all circles have a cloud cover below 0.4 for most instruments except

VELOX with 90 % cloud cover estimates below 0.6. Furthermore, about 50 % of the time cloud cover estimates are below 0.2. The analysis of circle-mean cloud cover suggests a high abundance of low cloud cover situations. WALES as well as the passive instruments are capable of detecting the thinner cloud edges and small and optically thin clouds. The HAMP radar is not sensitive to these cloud parts which typically consist of small cloud droplets. The comparison illustrates the potential of the multi-instrument cloud cover product to study cloud macro- and microphysical properties.

## 4   Accessing EUREC⁴A's HALO data


EUREC⁴A was a large field campaign, which involved hundreds of people from dozens of institutions spread over a score of countries across three continents. Measurements were collected from more sensors than people. The task of quality controlling and curating the resultant data is immense and time consuming. Making the data visible and usable by a broader community is even more daunting, all the more so for those same qualities that made EUREC⁴A's execution so successful – namely the

multiplicity of people, institutions, and countries involved.

EUREC⁴A's HALO is a microcosm encapsulating many of the challenges faced by EUREC⁴A as a whole. HALO deployed instruments developed and operated by different groups, funded by different agencies, and designed to collect very different types of data. Fig. 9, graphically illustrates many of these relationships. Synchronizing the processing, release, and even archiving of this data is neither practical nor desirable. Instead, to make HALO data visible and more readily usable, also as new

data-products are published and released, a few of the present authors created an online book. Initially the book collected and distributed use cases as a form of "How to" that others could follow. This approach to data dissemination caught on within in the EUREC⁴A community, and investigators from other platforms added their own use cases. This lead to the development of "How to EUREC⁴A", an online and interactive Jupyter book. "How to EUREC⁴A" is now hosted on the EUREC⁴A domain[1] and serves as the recommended entry point for those interested in accessing and using HALO data.

The chapters of "How to EUREC⁴A" are built from a combination of code and explanatory markdown files. The use cases range from simple examples that show how to work with HALO flight segments, to simple quick-looks of data from an individual instrument, to more elaborate analyses that combine measurements from different instruments. For example, the comparison of cloud cover shown in Figure 6 is one of the use cases documented in the book. Code examples are written in Python, but the methods employed are readily transferred to other languages – even by those unfamiliar with Python.

All example scripts can be run interactively in the browser via a binder integration such that no local setup and memory resources are necessary. The code examples can also be downloaded and run locally with the respective requirements for the python environment installed. "How to EUREC⁴A" thus provides a common starting point for those interested in working with EUREC⁴A's HALO data, and at the same time serves as a tutorial to help inexperienced users begin using the data.

---

[1] https://howto.eurec4a.eu

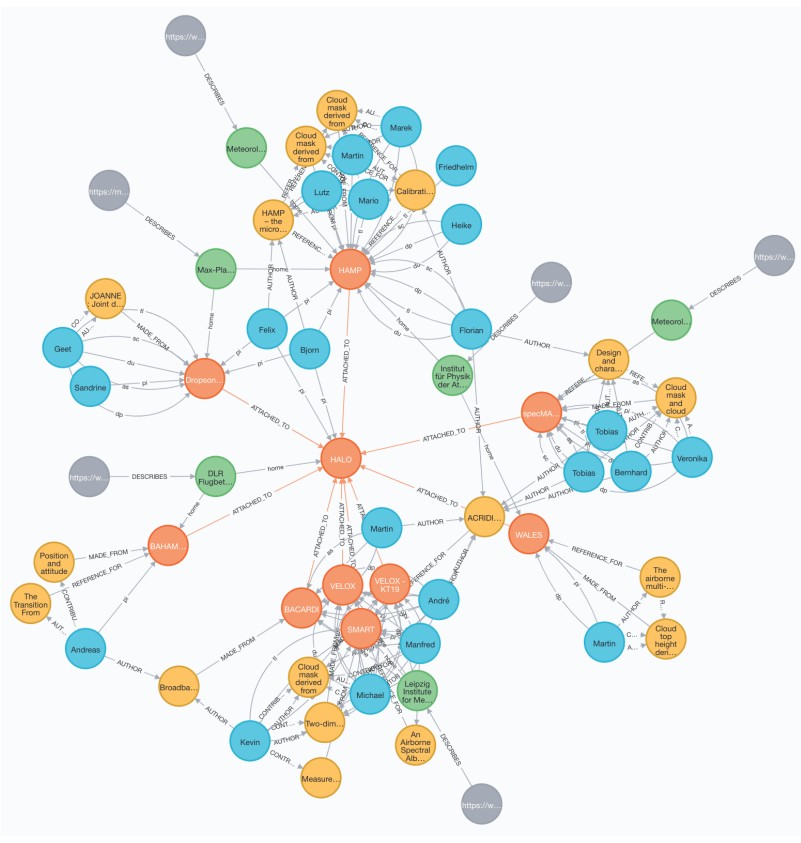

**Figure 9.** Graphical representation of the instrument payload information that is provided in the "How to EUREC⁴A" online book and the instrument information file.

"How to EUREC⁴A" is a living document. It continues to mature through the addition of chapters on new instrument plat-

forms, through the addition of new or corrections of old analyses, and through the ingestion of new data, or data releases, and their provenance. For this latter purpose, and to help disassociate the indexing of data from its archiving, "How to EUREC⁴A" accesses EUREC⁴A data through an intake catalog[2], which is continuously updated to contain links to the most recent versions of the publicly available EUREC⁴A data. To provide users with a more narrative description of the available HALO data, "How to EUREC⁴A" also contains a section that links the HALO scientific payload with its institutional owners, contact

information to its data providers, citations to reference material, and links to data. For users concerned about the volatility of an online book, a snapshot of this information, valid at the time of submission, has been compiled into a machine readable YAML (YAML ain't markup language) file (Kölling et al., 2021), it includes the information shown graphically in Fig. 9 and is included as an immutable asset with this paper.

---

[2]https://github.com/eurec4a/eurec4a-intake



## 5   Summary

We describe the operation of the German research aircraft HALO during the EUREC[4]A experiment, and its associated data. HALO flew fifteen scientific missions during EUREC[4]A. These are described both by the scientific measurements made from instruments aboard the aircraft but also through the provision of auxiliary data: time-stamp intervals that segment the flight paths; selected and curated photographs from each flight; movies showing the evolving satellite presentation and flight tracks for each flight; sub-setted aircraft state data; and cloud masks from instruments sensitive to the presence of clouds.

In addition, meta data is provided describing deployed instruments, their particular configuration, contact information to those responsible for each instrument, and its data, and (when available) a URL to the data itself, is included, along with the aforementioned auxiliary data, through a machine readable text (YAML) file. For convenience, Table 6 provides links to all of the data assets publisehd with this manuscript. In addition, "How to EUREC[4]A" provides a much more flexible and comprehensive, but for now volatile, description of EUREC[4]A's HALO's data. This book, first developed for HALO, is (as the

name suggests) being extended to other instrument platforms deployed during EUREC[4]A. It is also being adopted by planned HALO campaigns. We hope that it will contribute to a new foundation for the treatment and dissemination of Earth-system science data.

**Table 6.** EUREC[4]A's HALO data

| Data set | Link | Citation |
| --- | --- | --- |
| Flight segments | https://doi.org/10.5281/zenodo.4900003 | Prange et al. (2021) |
| Satellite movies | https://doi.org/10.25326/225 | Schulz et al. (2021) |
| Photographs | https://doi.org/10.25326/229 | Konow et al. (2021) |
| Instrument information | https://doi.org/10.25326/232 | Kölling et al. (2021) |
| Aircraft state | https://doi.org/10.25326/161 | Klingebiel (2021) |
| HAMP radiometer cloud mask | https://doi.org/10.25326/223 | Jacob (2021a) |
| HAMP radar cloud mask | https://doi.org/10.25326/222 | Jacob (2021b) |
| specMACS cloud mask | https://doi.org/10.25326/166 | Pörtge et al. (2021) |
| VELOX KT19 cloud mask | https://doi.org/10.25326/162 | Schäfer et al. (2021a) |
| VELOX IR imager cloud mask | https://doi.org/10.25326/163 | Schäfer et al. (2021b) |
| WALES cloud mask | https://doi.org/10.25326/216 | Wirth (2021) |
| "How to EUREC[4]A" | https://howto.eurec4a.eu | |

## 6   Code and data availability

Code and data are freely available at the locations specified in Table 6.





## Appendix A: Data set updates

Several data sets recorded with HALO during EUREC[4]A continue and extend data sets that were published in earlier publications like Konow et al. (2019). These extensions, which are published alongside this paper, are described in the following appendix. Detailed previous data publications are referenced for the main processing steps and novelties or differences in the context of EUREC[4]A are pointed out.

### A1 HAMP microwave radiometer brightness temperatures

HAMP microwave radiometer brightness temperatures observed during EUREC[4]A extend the data set of previous campaigns such as NARVAL and NARVAL2 which are published by (Konow et al., 2019). Compared to previous campaigns a technical update of the radiometers and their data acquisition system resulted in a reduction of the number of frequency channels from 26 to 25, i.e. the $183.3 \pm 12.5$ GHz was omitted.

The quality of the original brightness temperature measurements was evaluated by comparing them with synthetic ones simulated from dropsonde profiles provided in the JOANNE data set considering the suggested humidity correction (George et al., 2021). Systematic differences which can arise from insufficient pre-flight calibration are corrected using a linear relation between recorded and synthetic brightness temperatures, which is estimated for each flight and radiometer channel. The high number of dropsondes released during each EUREC[4]A flight allowed the implementation of a linear correction as an update from the previous processing (Konow et al., 2019), which used a simple offset.

The radiometer data was recorded on three independent data acquisition computers. The clocks of all systems were configured such that they occasionally synchronize with the central HALO BAHAMAS system clock. However, when inspecting the time series of different recording computers, clear time offsets in the order of seconds can be identified. To correct for these time offsets, the brightness temperature time series were carefully inspected and time series of the different modules were compared with each other, the WALES data and the radar. Doing so, we could identify offsets between -7 and +2 s. with the exception of one modules clock running 141 s behind during the flight HALO-0119 which were subsequently corrected.

In addition, we did a manual inspection of the brightness temperature time series for non-atmospheric signals. This means that signals coming for example from thermal receiver instabilities and emission signals observed over transient objects like ships, which have a much higher microwave emissivity than the ocean, are discarded. The brightness temperature and time offsets are corrected. Due to the new data acquisition system, data with a high temporal resolution of 4 Hz sampling rate are available on request in addition to the quality controlled and published data set.

### A2 HAMP microwave radiometer retrievals

The retrieval methods developed by Jacob et al. (2019) are applied to the EUREC[4]A observations and the retrieved time series of integrated water vapor, liquid water path and rain water path are published in Jacob (2021c). For this data set, we updated the training database for the artificial neural network retrieval using ICON simulations for the EUREC[4]A period.



## A3 HAMP cloud radar calibration

The absolute calibration of radar reflectivity measured by the HAMP cloud radar followed Ewald et al. (2019). This technique uses the well-defined ocean surface backscatter as external calibration reference. For that purpose, the angular ocean surface backscatter was sampled several times using dedicated flight maneuvers (as described in the flight segmentation data). In

total, six maneuvers with a constant roll angle of $10°$ and alternating roll maneuvers of $\pm20°$, identified as segments 'radar calibration tilted' and 'radar calibration wiggle', respectively (Sect. 2.2), were performed during EUREC[4]A.

**Table A1.** Absolute calibration offsets $\Delta\sigma_0$ found in comparison with the ocean surface backscatter $\sigma_0$ for the HAMP cloud radar during EUREC[4]A. Furthermore, the horizontal wind speed $u_{\text{drop}}$ measured by dropsondes are compared with the horizontal wind speed $u_{\text{fit}}$ retrieved from the angular pattern of $\sigma_0$.

| Date | $\Delta\sigma_0$ [dB] | $u_{\text{fit}}$ [m s$^{-1}$] | $u_{\text{drop}}$ [m s$^{-1}$] |
|---|---|---|---|
| 2020-01-28 | +1.74 | 6.0 | 6.3 |
| 2020-02-02 | +1.77 | 5.8 | 3.2 |
| 2020-02-07 | +1.51 | 9.1 | 11.5 |
| 2020-02-09 | +1.67 | 8.4 | 11.8 |
| 2020-02-11 | +1.67 | 8.3 | 12.5 |
| 2020-02-13 | +1.59 | 7.7 | 10.9 |
| avg. | +1.7 | $\Delta u = -1.8\,\text{m s}^{-1}$ | |

Based on measured signal-to-noise ratios, the normalized radar cross section $\sigma_0$ was calculated using the system parameters listed in Ewald et al. (2019). Due to a receiver update and a frequency change from $35.5$ to $35.17\,\text{GHz}$, the antenna gain ($G_a = 49.0\,\text{dBi}$) and the receiver noise figure (NF $= 8.4\,\text{dB}$) had to be redetermined in laboratory measurements. After correcting

measured $\sigma_0^*$ for gaseous attenuation, they could be compared to modeled $\sigma_0$ using horizontal wind speed data from collocated dropsonde soundings analogous to Ewald et al. (2019). In Table A1 the absolute calibration offsets found $\Delta\sigma_0 = \sigma_0^* - \sigma_0$ are summarized for each successful calibration pattern. In addition, Tab. A1 compares the horizontal wind speed $u_{\text{drop}}$ measured by the dropsondes with the horizontal wind speed $u_{\text{fit}}$ retrieved from the angular pattern of $\sigma_0^*$. In summary, an absolute calibration offset of $+1.7\,\text{dB}$ was found for the EUREC[4]A deployment of the HAMP cloud radar and subsequently subtracted from the

radar reflectivity for the unified dataset.

## Appendix B: Cloud Masks

In the following the methods used to construct cloud masks for each instrument are detailed.



## B1   WALES

For the WALES cloud mask lidar raw data at a temporal resolution of 5 Hz were used, which corresponds to 40 m horizontal
spacing. The vertical resolution of the backscatter data is 7.5 m. The cloud flag is inferred from a step of the lidar backscatter
ratio at 532 nm to values bigger than 10, while searching from the aircraft downwards. The limit value of 10 is higher than what
is expected from dry aerosol at this time of the season and thus indicates that a considerable water uptake has occurred. To
further facilitate the discrimination of optically thin clouds and opaque ones the atmospheric optical depth between the cloud
top and the sub-cloud layer is included in the data set. Values above 2 to 3, depending on the background light situation, are
only rough estimates, but indicate the presence of on optically thick, opaque cloud. The details of this method, which is based
on the HSRL channel, can be found in Esselborn et al. (2008). The data also includes the altitude of the cloud top as height
above the EGM96 geoid with a precision of about 10 m and an accuracy of about the same magnitude. Further included is an
estimation of the altitude of the top of the boundary layer above sea level. This is inferred from the maximum correlation of
the lidar backscatter ratio profile at 532 nm with a step function. This quantity is experimental and should be interpreted with
care, especially in the presence of residual layers or strong horizontal wind shear which may cause multi-layer structures. In
the case of a cloud, no PBL-top is given. To enable precise comparisons with other instruments the WALES cloud-top data
also includes the position of the target-cloud in addition to the coordinates of the aircraft. These two locations may differ by
several kilometers depending on roll angle. This cloud mask data set is published (Wirth, 2021) and is available for download
under https://doi.org/10.25326/216.

## B2   HAMP (cloud radar)

The HAMP radar cloud mask uses radar reflectivities measured by the HAMP cloud radar and calibrated as described in
Appendix A3. The data is provided at a 1 Hz time interval and with 30 m vertical resolution. Reflectivity data are first filtered for
clutter. Any signal above the noise level at 200 m above sea level or higher, is considered a possible cloud. Signals originating
from object of at least four contiguous pixels are classified as *most likely* a cloud, otherwise it is *probably* a cloud signal. The
cloud mask is published by Jacob (2021b) and can be downloaded from the database (https://doi.org/10.25326/222).

## B3   specMACS

The data of the shortwave-infrared line camera of specMACS at a temporal resolution of 30 Hz is used to provide a cloud
mask. The cloud mask is based on two criteria: the brightness of the observed pixels and the strength of absorption due to
water vapor. For evaluating a scene two reference spectra in the range from 1015 nm to 1300 nm (one with and the other
without molecular absorption - abbreviated as $L_{\text{abs},\lambda}$ and $L_{\text{no abs},\lambda}$) are calculated. The simulated transmittance $T_{\text{ref},\lambda}$ is then
given by $L_{\text{abs},\lambda}/L_{\text{no abs},\lambda}$. These reference spectra are fitted to the measured radiances ($L_{\text{meas},\lambda}$) using the following equation:

$$L_{\text{meas},\lambda} = a\, L_{\text{no abs},\lambda}\, (T_{\text{ref},\lambda})^x$$



The fit parameter $a$ scales with the brightness of the measurements and the parameter $x$ is a measure of absorption. Two different thresholds are applied to the brightness fit parameter to discriminate between *most likely cloudy* (if the brightness of a

pixel is higher than the upper threshold), *probably cloudy* (if the brightness is between both thresholds) and *cloud free* pixels (if the brightness is smaller than the lower threshold). This brightness criterion is not sufficient for ocean areas influenced by sunglint, which can be misclassified as cloudy due to the bright glint. To address this case sun-glint situations are first identified by theoretical considerations depending on solar illumination and viewing geometry. Because the near-surface abundance of water vapor results in a much larger water-vapor path length in cloud free versus cloudy scenes, the latter can be distinguished

from the former by the water vapor absorption. The absorption is derived from the measurements using the second fit parameter $x$. The threshold for this fit parameter is initially derived by visual inspection of a reference scene. Afterwards it is adapted dynamically depending on the viewing zenith angle of the camera, the solar zenith angle, and the column integrated water vapor density of ECMWF ERA5 reanalysis data. If sun-glint is present this method is applied to all pixels classified as either probably or most likely cloudy. Pixels with a strong vapor absorption signal are set to *cloud free*. The cloud mask dataset has

been published (Pörtge et al., 2021) and is accessible for download (https://doi.org/10.25326/166).

### B4   VELOX (IR Imager)

A two-dimensional cloud mask from VELOX has been derived with a similar method using brightness temperature measurements from the broadband channel of the instrument at a temporal resolution of 1 Hz. In this case, four thresholds (0.5 K, 1.0 K, 1.5 K, and 2.0 K) are used to determine a cloud mask flag for each spatial pixel in the field. As with the KT19 cloud mask, if

one threshold is exceeded, the pixel is flagged as *probably cloudy*, whereas all thresholds must be exceeded for the pixel to receive a *most likely cloudy* flag. Furthermore, a maximum and minimum possible cloud cover within the field of view has been calculated for each time step based on the number of probably cloudy and most likely cloudy pixels, respectively. This cloud mask data set has also been published (Schäfer et al., 2021b) and is available for download (https://doi.org/10.25326/163).

### B5   VELOX (KT19)

The cloud mask from the KT19 is derived by comparing the measured brightness temperature to simulated measurements in cloud-free conditions. Using three thresholds (0.7 K, 1.0 K, and 2.0 K) based on the difference between the measurements and simulations, a measurement is flagged as cloud-free when no threshold is reached, *probably cloudy* when one threshold is reached or *most likely cloudy* when all three thresholds are reached. The cloud mask from the KT19 has been published (Schäfer et al., 2021a) and can be found in the database (https://doi.org/10.25326/162).

### B6   HAMP (microwave radiometer)


The HAMP Microwave Radiometer cloud mask is based on thresholding liquid water path (LWP) retrievals. The LWP retrieval is based on the warm microwave emission signal by the clouds over the radiatively cold ocean surface as described by Jacob et al. (2019). Differences with respect to EUREC[4]A observations compared to Jacob et al. (2019) are explained in Section A2.



The LWP observations have a 1 s temporal resolution and are representative for a footprint of about 1 km. A scene is considered

*probably* and *most likely* cloudy if the LWP exceeds $20\,\mathrm{g\,m^{-2}}$, respectively, $30\,\mathrm{g\,m^{-2}}$. These thresholds correspond to (about) two to three times the clear-sky retrieval uncertainty, respectively. The clear-sky LWP offset correction (Jacob et al., 2019), which considers the WALES cloud mask and would allow for even lower thresholds, is not utilized here in order to provide a cloud mask that is independent of the other cloud mask products. The cloud mask is published by Jacob (2021a) and can be downloaded from the database (https://doi.org/10.25326/223).

*Author contributions.* FA, SB, GG, SG, MJ, BM, MM, SS, BS, JV, RV, and MWe acted as principal investigators on one or more research flights. WALES operation and data quality control was done by SG, MG, and MWi. The SMART, VELOX, BACARDI team who took care of instrument design, operation, and/or data quality control, is made up of AE, AL, MS, MWe, KW, and MZ. FA, SC, FE, GG, MH, LH, MJ, FJ, MK, HK, MM, JR, HS, and AW made up the HAMP team with different responsibilities for instrument design and improvement, instrument calibration, instrument operation, data quality control, and product generation. LF, TK, BM, VP, and TZ operated the specMACS

instrument and ensured data quality control and product generation. AG and MZ operated the BAHAMAS system and took care of data quality control. FE updated the HAMP radar calibration for EUREC4A. MJ and AW derived the HAMP radiometer calibration data. VP coordinated the joint cloud mask product. MJ, FG, VP, MS, and MWi derived the cloud mask products for their instruments. MP derived the original flight segmentation. Flight segments were manually identified by GG, LH, TK, HK, TL, TM, and MP. HS created dataset satellite images and movies. RV and HK organised photo collection and publication. MK created and published the BAHAMAS data subset. TM,

JR, and TK created the "How to EUREC4A" book. SB and BS devised and coordinated the EUREC4A campaign. FA led the participation of the German Research Foundation (DFG) partners. HK and BS devised and wrote the manuscript with text input from FE, GG, MJ, MK, TK, AL, TM, VP, JR, MS, HS, and MWi. All coauthors contributed with their ideas and comments to the development of the manuscript.

*Competing interests.* The authors declare that they have no competing interests.

*Acknowledgements.* This work was supported by the Max Planck Society and the DFG HALO SPP 1294. SB, JV and RV have received

funding from the European Research Council (ERC) under the European Union's Horizon 2020 research and innovation programme (grant agreement No 694768). We would like to thank Vincent Douet from AERIS and the AERIS team for their support in publishing the various data sets associated with this paper. The HALO flights would not have been possible without the support and work of the flight operations team. Thanks to the entire team for their support in planning and execution of these flights.





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
