# Peer review of "EUREC[4]A's HALO"

_Earth System Science Data, 2021_

## Author Comment (AC1)

**Authors' Response**

We would like to thank all reviewers for their helpful comments on the manuscript. The detailed questions and suggestions helped improving the manuscript a lot.

In the following, we will answer the comments of all reviewers in one document, starting with reviewer 1. Italics indicate reviewer comments and changes to the manuscript are written in grey.

**Comments from Reviewer #1**

*The text on Figure 9 is microscopic and unreadable. If all of the text is important, it should be made large enough to read. If the connector line text is not important, it should be removed, and perhaps only the text in the circles retained and made large enough to read.*

We thank Reviewer 1 for pointing this out. The purpose of Figure 9 is to show that HALO instrumentation is not run by only one institution, but instead is collaboratively built and operated by an interconnected community. Indeed, the text included in the figure was hard to read and probably not essential as that information is also included in the supplied YAML file. We have therefore made another figure, which omits the text and is more focused on the connections.

*L. 110: Principal, not principle.*

We have corrected this point.

**Comments from Reviewer #2**

*My first remark concerns the labelling of the cases, as proposed in Table 1, column "comments". Considering that the dataset is destined to become a permanent entry in the scientific literature, these funny names lose their interest. Replace them with a scientific summary of the case such as "coldPools", if a naming is deemed necessary. Later in line 65 it is mentioned that the names should "remind the reader of the principle investigator (PI)". This is another option, but then the names should be spelled out correctly, as later generations will need this information.*

We have been going back and forth on this topic as well. There are good arguments for keeping the names and for removing them. We have now changed the column title to "Informal moniker" instead of "comments" to clarify that this is extra information and not the official name of each flight. We agree with the reviewer comment that this moniker will lose its informational value in the coming years. Nevertheless, we would like to keep this column as a way of quick recognition for the people that were part of the campaign and remember the specifics of the discussions around individual flights. To refer to a specific flight we always use the Flight ID from the first column of Table 1.

*Another confusing description concerns the segments, that are called "kind" in line 71+. Is "type" a better wording? Include the term in Table 2 (segment type?).*

Thank you for raising this concern. We also thought about the naming a bit. According to the Merriam-Webster dictionary, the (first) definition of "kind" is: "a group united by common traits or interests". The (first) definition of "type" is: "a particular kind, class, or group". Accordingly, describing a particular kind or a type should be pretty much interchangeable. But probably, "type" has the tendency to be more definitive than "kind", in the sense that something usually is of one specific type, but can be part of many kinds. Regarding the segments, the "kind" is intended to be a form of grouping to be used when selecting a group of segments united by some common properties. One segment can belong to many such grouping criteria, if the criteria are themselves not exclusive. Thus,

a segment can be part of various kinds. As mentioned above, to us it sounds better to have multiple "kinds" associated with one segment than to have multiple "types" associated with one segment. Furthermore, "type" is more commonly used as a reserved word in various programming languages, so the use of "kind" can be simpler in some circumstances. We therefore concluded that "kind" might be better suited for this segmentation.

*In line 83 is written: "Segments also contain a field called 'dropsondes',". Where? I don't understand this sentence? Table 1 is referring to dropsondes, right?*

Thank you for pointing out that this sentence might be misleading. We here refer to the flight segments in the associated YAML files. There, for all segments during which dropsondes were deployed, the respective dropsondes are listed with their IDs. This helps identifying relevant dropsonde data for specific flight segments. We have changed the sentence to:

"The flight segments in the YAML files also contain a field called 'dropsondes', which provides a list of the dropsondes, whose time of launch are associated with the respective segment."

*Line 110+ Why do you insist who took a photo? Is it for copyright reasons? Otherwise, I find this information irrelevant.*

There are multiple reasons for listing the photographer as part of the picture meta data. First of all, yes, for copyright reasons. But the second, and probably more relevant, reason is so that users have an idea who to ask if they have questions regarding the pictures in the future. Therefore, we would leave the photographer information in the meta data.

*Figure 4: "The photographs are representative for the Fish, Flowers, Gravel and Sugar type of organization patterns" Explain!*

We have changed the wording in the figure caption to state a bit more clearly that the names, Fish, Flowers, Gravel and Sugar stem from the classification for trade wind organizational patterns that was developed by Stevens et al. 2020. The caption is now:

"Example photographs taken on board HALO with added meta data. The photographs are representative for the typical trade wind organizational patterns identified by Stevens et al. (2020) Fish, Flowers, Gravel and Sugar (from top left to bottom right)"

*Line 177: "measurements suggest higher cloud cover in the beginning as well as towards the end of the campaign" Why do you insist on averaging them if you know that they show a tendency. Better discuss the actual observed tendency.*

We would like to thank the reviewer for pointing out that this has not been clearly stated in the text. We do not average cloud cover observations from different flights. And we only use Figure 7 to give an overview of cloud cover observations from different instruments during the course of the campaign. We rewrote the paragraph to:

"To provide context to the variations in cloud cover, Fig. 7 shows a time series of circle-mean (minimum) cloud-cover estimates for all research flights and from all instruments respectively. HALO typically flew six circles per research flight (per day). In addition to the research flight mean, the whiskers span the range from the smallest to the largest circle-mean (minimum) cloud cover for each flight. In general, the individual instruments show a tendency for higher cloud cover at the beginning as well as towards the end of the campaign. For most cases the cloud cover estimates from passive instruments and the radar agree well. WALES systematically detects more clouds. It is more aligned with the circle-mean (maximum) cloud-cover estimates of the other instruments, as it does not include an uncertain cloud flag and is very sensitive to optically thin clouds. The flight HALO-0215 is an exception to the systematic difference between WALES and the other sensors which is due to a deep

stratocumulus layer with a strong reflection at cloud top that blinded the lidar, while the radar was still able to provide reasonable estimates."

*Line 209: "Synchronizing the processing, release, and even archiving of this data is neither practical nor desirable." I don't agree! This is highly desirable.*

Thank you for encouraging us to think about this topic again. We still would state that it is not desirable to synchronize processing, release and archiving of all data from all instruments of a campaign. We are encouraging all groups and participants to push their analysis and research forward independently. We don't want some individuals to block the progress of everyone else, potentially delaying the release of data indefinitely. Thus, we don't want to synchronize processing, releasing or archiving, but want to enable asynchronous schedules. Having a common archive is potentially desirable, but it's currently impractical as there's currently no single archive known to the authors which is willing to accept the entire dataset taken during EUREC4A. Furthermore, the dependence on a single archive seems to be unfortunate. We are working on technologies which will enable a distributed archive with a common interface across datasets, but that is still an ongoing development. In the meantime, we have to rely on readily available technologies (i.e. HTTP, DOI etc...).

*Which brings me to my main concern: the "How to Eurec4A"*
*""How to EUREC4A", an online and interactive Jupyter (what is that???) book"; "*

Thanks for pointing out that this term is not immediately recognizable to everyone. We have added a footnote explaining the term Jupyter to the manuscript. Jupyter is an interactive development environment that can be used with several programming languages.

*"How to EUREC4A" is a living document. It continues to mature through the addition of chapters",…..*
*Table 6 gives the DOI of the data from the different instruments. Those seem to be frozen in time by the DOI. However, this "book" seems to treat the data in an evolving way and will be complemented by new campaigns and more data, different procedures, …*
*From the point of view of a future generation scientist that wants to exploit the campaign data referenced in this paper and understand how they were combined to reach the published conclusions, this will become an impossible task. The current protocol will be lost in the various modifications made since.*
*Thus, I would advocate to freeze a copy of the current state of the "book" and DOI it, in order to conserve the current combination of different results for future reference.*
*The "How to Eurec4A" is the main new aspect, in addition to the catalogue of the already published other datasets. Thus, a correct preservation would seem a mandatory condition for acceptance of the current paper.*

The purpose of the submitted manuscript primarily is to describe and reference the published datasets whose DOIs are listed in Table 6. The citations that are added in the third column of Table 6 are citations of the data sets themselves but not of other publications. Therefore, the main data sets that are published with this paper are the referenced data sets.

While we like the "How to EUREC4A" book a lot, it is merely an additional tool we provide for the data. The main aim of the "How to EUREC4A" book is to enable and teach the direct use of datasets from the EUREC4A field campaign. This is important if multiple datasets are to be combined and if analysis scripts should be shared among scientists. We generally want to encourage to combine multiple datasets and share analyses among colleagues.

Ideally, a system which enables this usage pattern should also be archivable (i.e. by freezing and keeping a version of the book) and thus be preserved for future reference, just as you suggest. There is however a catch: this desired usage pattern in a frozen "How to EUREC4A" book would require some function which accepts a dataset identifier and returns the identified dataset unchanged and

independent of the local computing environment. Currently, there is no such mechanism which does this in a stable, future-proof way and which is commonly used in our field.

A DOI provides us a means to manually find and retrieve some (probably) frozen dataset. (probably, because DOIs provide no hard guarantees about how warm a frozen dataset may become...). Table 6 is in support of this access mode, as Table 6 will also be frozen with this manuscript and is manually readable as well. However, this does not help in the context of the "How to EUREC4A" book, which requires automatic retrieval of the referenced data. Admittedly, it sounds a bit ridiculous, but as long as there is no mechanism which enables stable references directly to a dataset (not to a landing page as it is currently the case for DOIs), we will have no choice but to manually update a catalog of storage locations of the frozen datasets whenever they change. Otherwise, the frozen book would stop working with the likewise frozen datasets in the near future. Furthermore, if there will be such a mechanism in future, we want to use this mechanism in a future version of the book, such that we don't have to update it manually for all time.

So, yes, we do in principle want to make datasets and usage frozen for future reference. Table 6, the referenced data DOIs and the analyses within the manuscript do this for the classic, manual use-case. Due to the reasoning above, a non-frozen book will make the automated analyses more available for the future than a frozen one and thus, we consider a reference to the most current version of the book more useful and with a higher chance for being in a functioning state than a frozen copy. As a way to conserve the state of the book that is representative of the point in time when this manuscript was created, we have added a copy of the current state of the "How to EUREC4A" book as a supplement to this manuscript.

*Minor remarks:*

*Line 15: add references for NARVAL-South and NARVAL2*

We have added references for these two campaigns.

*Line 24: what is the meaning of "looser coordination"?*

We have rewritten parts of this paragraph to explain this term a bit more. It now reads:

HALO was one of four scientific platforms forming the nucleus of EUREC4A. Its measurements were closely coordinated with those from the other three core platforms – the research vessel (R/V) Meteor, the Barbados Cloud Observatory (Stevens et al. 2016), and the French SAFIRE ATR-42 – to facilitate observations of the same air mass from different vantage points. Two additional aircraft, three further research vessels and a small fleet of air- and water-borne robotic instrument platforms supported a substantial broadening of EUREC4A's initial scope and, as described by (Stevens et al. 2021), involved looser coordination with HALO. Often day-to-day operation time and area of these platforms were not closely matched with the platforms mentioned above, but these platforms were still operated in the general EUREC4A area during the campaign period and coordinated measurements were still conducted from time to time.

*Line 25: "We do so by **by** describing how HALO was **tasked** during" replace by "We do so by describing how HALO was deployed during"*

We have changed this.

*Line 41: "document the meteorological conditions (through photographs and". Add "also" instead of bracket, as I imagine that other data were also used to characterize the meteorological conditions apart from photos*

We have rephrased this sentence to:

This description is aided by the development of a meta data concept (and the meta data arising from its application) to systematically segment the flight data and document the meteorological conditions (amongst others through photographs and satellite imagery) encountered on the different flights.

*Table 1 caption: "all flights were local flights in that they took" replace by "all flights were local flights that took"*

We have changed this.

*Line 53: explain BCO*

We have added the explanation of the abbreviation and the citation here as well.

*Line 70: replace by "the second denoting the end of the segment"*

We have rephrased this sentence a bit to:

" […] the second denoting the first time step after the end of the segment"

The second time step of a segment does not denote the last time step of this segment but the first time step that is outside this segment. Therefore, we kept the original phrasing but added the word "step" in hopes of clarifying this matter a bit more.

*Line 80: delete "or avoidable deviations from the kind definitions" or explain better*

We have rephrased this part to:

"least one other person later tested the segmentation for errors or avoidable deviations from the segment definitions (Tab. 2). This was done to maintain the objective segment classification whenever possible so that the user of this data can expect the segments to match the definition as closely as possible."

*Line 81: replace "Because" with "As"*

We have changed this.

*Line 130: replace "Information on to how" by "Information on how"*

We have changed this.

*Line 194: "Viewed differently, about 90%". I don't understand the meaning of the expression. Please explain.*

We have rephrased this sentence to:

Or, to put it another way, about 90% of all circles have a cloud cover below 0.4 for most instruments except VELOX with 90 % cloud cover estimates below 0.6.

*Line 227: replace "intake" by "input"*

We have decided to keep the word "intake" in this case since it is the name of a software package that is widely used. That way, users not familiar with the EUREC4A data, but familiar with the concept of an intake catalogue will be able to quickly recognize the concept behind this data catalogue.

Comments from Reviewer #3

1. *KT19 is shown in Fig. 6, its abbreviation should be given earlier, already in Table 4.*

Thank you for pointing this out. We have added the remark ", abbreviated KT19 in this paper" to Table 4

2. *Fig. 1: Barbados is somewhat hidden, better to mark it clearer on the map.*

We have adapted this Figure so that Barbados now is more visible on the map.

3. *Fig. 3 caption: It is unclear what the authors refer to as "ferry flights".*

We have changed the caption to:

"Snapshots are from about mid-flight time of HALO, except for the transfer flights to and from Barbados (HALO-0119 and HALO-0218)."

4. *Fig. 9 is unreadable with this size. Please provide details as to the meaning of the colors and make the font (much) larger. Can also provide a more readable online version if the details are not critical here, but if the figure should appear in the print version, it is needed to provide more details on the general aspects shown.*

Thank you for pointing this out. We decided to replace Figure 9 with a version that omits the text and is more focused on the connections. This way, we hope to still convey the message that that HALO instrumentation is not run by only one institution, but instead is collaboratively built and operated by an interconnected community

5. *Table 6: Are the dropsonde data accessed under "flight segments" please indicate explicitly where.*

The dropsonde data are published together with another manuscript by George et al., (2021). We did not list the data in Table 6 because they are not part of the assets published together with this manuscript and we would like to avoid the impression that they were.

We have added some words of clarification to the Summary:

"Not all data sets from the instruments that were onboard HALO are assets to this manuscript. Most of the data sets are published separately (e.g. the dropsonde data set JOANNE by George et al. (2021)). We only describe a subset of all observed data that were taken on HALO during EUREC4."

*Typos:*

1. *Line 22: change "aircraft" to "aircrafts"*

According to Meriam Webster and the Oxford English Dictionary, the plural of the word "aircraft" is also "aircraft" (OED marks the word "aircrafts" for the plural as rare). We therefore decided to keep the word aircraft throughout the manuscript even if it refers to the plural.

2. *Line 211: delete "in"*

We have changed this. Thank you.